# Diagnosis of Duchenne Muscular Dystrophy in a Presymptomatic Infant Using Next-Generation Sequencing and Chromosomal Microarray Analysis: A Case Report

**DOI:** 10.3390/children8050377

**Published:** 2021-05-11

**Authors:** Eun-Woo Park, Ye-Jee Shim, Jung-Sook Ha, Jin-Hong Shin, Soyoung Lee, Jang-Hyuk Cho

**Affiliations:** 1Department of Rehabilitation Medicine, Keimyung University Dongsan Hospital, Keimyung University school of Medicine, Daegu 42601, Korea; parkew1234@naver.com (E.W.P.); soyounglee3@gmail.com (S.L.); 2Department of Pediatrics, Keimyung University Dongsan Hospital, Keimyung University school of Medicine, Daegu 42601, Korea; dapdapgirl@hanmail.net; 3Department of Laboratory Medicine, Keimyung University Dongsan Hospital, Keimyung University School of Medicine, Daegu 42601, Korea; ksksmom@naver.com; 4Department of Neurology, Pusan National University Yangsan Hospital, Pusan National University School of Medicine, Yangsan 50612, Korea; shinzh@gmail.com

**Keywords:** muscular dystrophy, Duchenne, genetic testing, creatinine kinase, early diagnosis

## Abstract

Duchenne muscular dystrophy is a progressive and lethal X-linked recessive neuromuscular disease caused by mutations in the dystrophin gene. It has a high rate of diagnostic delay; early diagnosis and treatment are often not possible due to delayed recognition of muscle weakness and lack of effective treatments. Current treatments based on genetic therapy can improve clinical results, but treatment must begin as early as possible before significant muscle damage. Therefore, early diagnosis and rehabilitation of Duchenne muscular dystrophy are needed before symptom aggravation. Creatine kinase is a diagnostic marker of neuromuscular disorders. Herein, the authors report a case of an infant patient with Duchenne muscular dystrophy with a highly elevated creatine kinase level but no obvious symptoms of muscle weakness. The patient was diagnosed with Duchenne muscular dystrophy via next-generation sequencing and chromosomal microarray analysis to identify possible inherited metabolic and neuromuscular diseases related to profound hyperCKemia. The patient is enrolled in a rehabilitation program and awaits the approval of the genetic treatment in Korea. This is the first report of an infantile presymptomatic Duchenne muscular dystrophy diagnosis using next-generation sequencing and chromosomal microarray analysis.

## 1. Introduction

Duchenne muscular dystrophy (DMD) is a lethal and progressive X-linked recessive neuromuscular disease, the most common form of muscular dystrophy [1,2]. It is caused by mutations in the dystrophin gene, which results in absent or insufficient functional dystrophin [2]. DMD is usually diagnosed in early childhood after suggestive complaints are noticed between three and five years of age [1,2,3]. Due to initial concerns and a lack of effective treatments to improve outcomes, DMD is associated with a high rate of diagnostic delay. For these reasons, the importance of early diagnosis and treatment of DMD tends to be underestimated [4]. Currently available treatments based on exon-skipping technology can improve clinical outcomes, but the treatment must begin as early as possible before significant muscle damage [4,5,6]. Therefore, early diagnosis and rehabilitation before symptom progression in DMD are key to keeping the muscles supple and preventing or minimizing tightness at the joints.

Creatine kinase (CK), a diagnostic marker of suspicious muscular dystrophy, is highly elevated in neuromuscular disorders with rhabdomyolysis [5,6,7]. Although it is difficult to make a differential diagnosis of neuromuscular disorders in the absence of profound hyperCKemia, next-generation sequencing (NGS) has been shown to be effective for diagnosing neuromuscular disorders [8]. The use of NGS for diagnosing DMD also had recently been in the spotlight [9]. Chromosomal microarray analysis (CMA) can also be used to identify the underlying congenital causes of unexplained developmental delay [10]. To the authors’ knowledge, the diagnostic application of NGS and CMA for infantile presymptomatic DMD has not been previously reported. Herein, the authors present a case of an infant patient with DMD with high hyperCKemia diagnosed using NGS and CMA before clinical-symptom appearance.

## 2. Case Report

A febrile four-month-old male infant was presented to the emergency department of a tertiary hospital. He was born at 38 weeks and 5 days of gestational age, weighing 3100 g. According to the Korean National Health Screening Program for Infants and Children, which is conducted at four months of age, the infant’s growth and development was in the normal range. The parents stated that the infant was usually healthy, and no disease had been reported in the medical files. The infant had no obvious hypotonic feature. His traction response and ventral suspension were normal.

The infant’s body temperature, systolic and diastolic blood pressures and pulse rate were 38.8 °C, 80 and 50 mmHg, and 170 beats/min, respectively. His laboratory results were as follows: white blood cell count, 20,800/µL (reference: 4000–10,000/µL); hemoglobin, 12.00 g/dL (reference: 13–17 g/dL); platelets, 478,000/µL (reference: 130,000–400,000/µL); C-reactive protein, 2.6 mg/dL (reference: 0.0.0.5 mg/dL); erythrocyte segmentation rate, 17 mm/h (reference: 0–15 mm/h); aspartate aminotransferase (AST), 380 U/L (reference: 0–40 U/L); and alanine aminotransferase (ALT), 285 U/L (reference: 0–190 U/L). He was admitted to the hospital’s department of pediatrics, and conservative treatment was initiated while considering the possibility of hepatitis, acute otitis media, or other viral infections. On admission day 2, the fever subsided, and laboratory results relevant to infection were improved; however, his AST and ALT levels continued to increase. Abdominal ultrasonography for hepatic dysfunction was unremarkable. The influenza B virus (antigen and polymerase chain reaction) and muscle enzyme tests were also performed, as myositis caused by influenza B was prevalent at the time. Myositis induced by influenza B virus was excluded due to the negative results of the influenza B virus test, although the infant did have a highly elevated serum CK level of 19,058 U/L (reference: 0–190 U/L). The enzyme activities related to lysosomal storage diseases were also in the normal range. Echocardiography and troponin I levels for cardiac function were normal. Rhabdomyolysis induced by fever was also considered, but it was excluded due to normal kidney function.

Diagnostic exome sequencing (DES) and CMA were performed together to ensure an accurate diagnosis since the patient was so young and not yet showing clear signs. Before implementing the DES and CMA, we referred the case to a genetic counseling specialist of laboratory medicine to minimize possible ethical issues. We also explained the detailed process of these genetic diagnostic techniques to the patient’s parents before providing written informed consent. For DES, genomic deoxyribonucleic acid (DNA) was extracted from peripheral blood leukocytes using the CheMagic™ Magnetic Separation Module I method (PerkinElmer chemagen, Besweiler, Germany) with the genomic DNA Blood 200 µL Kit. The xGen Inherited Diseases Panel (Integrated genomic DNA Technologies, Inc., Coralville, IA, USA) was used for library preparation, and sequencing was performed on the Illumina NextSeq 500 platform (Illumina, Inc., San Diego, CA, USA), generating 2 × 150 bp paired-end reads. The obtained sequence reads were aligned to the hg19 human reference sequence using the Burrow–Wheeler Aligner (BWA version 0.7.12, http://bio-bwa.sourceforge.net, assessed on 17 June 20). Duplicated reads were removed with Picard tools (version 1.96, http://picard.sourceforge.net). Local realignment, recalibration, and variant calling were performed with the Genome Analysis Tool Kit (GATK version 3.5, Broad Institute, Cambridge, MA, USA) and annotation was performed with Variant Effect Predictor (VEP88), dbNSFP v3.3. Hemizygous deletion of exons 45–50 was detected (Figure 1A). For CMA, genomic DNA was extracted from the peripheral blood. CMA was performed with the CytoScan 750 K Array (Affymetrix, Santa Clara, CA, USA) according to the manufacturer’s recommendations. The platform was composed of 550,000 nonpolymorphic copy number variation probes and more than 200,000 single nucleotide polymorphism probes with an average resolution of 100 kb. All data were visualized and analyzed with the Chromosome Analysis Suite (ChAS, version 1.3.0.46, Life Technologies Ltd., 3 Fountain Drive, Paisley, Scotland, UK) software package (Affymetrix) using hg19 human genome build. As a result, the patient’s microarray profile showed a hemizygous deletion of ~240 kb (31795774_32036509) in the chromosome region Xp21.1 (Figure 1B).

The patient’s general condition improved, and he was discharged on the seventh day of hospitalization. He was diagnosed with DMD and is attending follow-up appointments at the hospital’s pediatric, neurology, and rehabilitation medicine departments. The patient is a candidate for exon-skipping therapy and is awaiting approval of this genetic treatment in Korea. He underwent a rehabilitation program that focused on gross motor function. A developmental evaluation performed at 10 months of age showed a delay of approximately 4 months in gross motor skills; his gross motor age using the gross motor function measure and Denver developmental screening test was 6–6.5 months. Afterward, the patient was transferred to the National Center for Rare Diseases in his region.

## 3. Discussion

DMD is caused by mutations in the dystrophin gene that result in absent or insufficient functional dystrophin, a cytoskeletal protein that enables the strength, stability, and function of myofibers [2]. The progression of DMD leads to weakness, associated motor delays, loss of ambulation, respiratory impairment, and cardiomyopathy [2]. In addition, comorbidities may include intellectual disability, delayed speech, and attention deficit disorder [3]. Usually, DMD is diagnosed in early childhood after observing the following signs: frequent falls, inability to run or climb stairs, difficulty in getting up from the floor, and Gowers’ sign [1,2,3].

Despite being a serious genetic disease, DMD is associated with a high rate of diagnostic delay, and effective early treatments are lacking. DMD diagnosis is reportedly delayed up to two years, as indicated by the period from onset of initial symptoms to the time of diagnosis [4]. However, this may be changing, as some studies recommend treatment with glucocorticoids when the diagnosis is recognized before substantial physical decline and early symptom onset [2,6]. As using exon-skipping technology may improve clinical outcomes, especially if initiated before symptom onset or before significant muscle loss has occurred [4,5,6], early diagnosis and rehabilitation are becoming even more important to keep these patients’ muscles supple and to prevent or minimize tightness at the joints, which will help sustain the ability to self-regenerate.

Serum transaminases, including ALT and AST, are included in the serologic panel routinely used to screen patients for evidence of liver disease as specific indicators. These transaminases are also present in high concentrations in the muscles [7]. Hypertransaminasemia can also be seen in patients with a muscle injury in the early stages before symptoms of muscle weakness are noted [7]. An elevated CK level may be seen with cardiac and skeletal muscle injury associated with muscle, nerve, and motor neuron disorders affecting the skeletal muscles [11]. HyperCKemia represents a nonspecific marker of muscle damage and can occur in these diseases, even when symptoms are not clinically apparent [11,12]. Therefore, serum CK samples should be obtained from patients with elevated ALT and AST levels to discriminate between muscle injury and liver disease, if necessary.

CK is highly increased with the disruption of the sarcolemma in the muscle and is considered a sensitive diagnostic marker of suspicious muscular dystrophy [5,6,7]. HyperCKemia in patients with DMD is characterized by an elevated CK level, that is, at least 10–20 (often 50–200) times the upper limit of the normal CK level [1]. Various studies have confirmed that children with DMD gene mutations had high hyperCKemia levels (≥2000 U/L) [4,5,13]. Infants with latent muscular dystrophy are found to have high elevations in the CK levels before symptoms manifest [12]. Overall, the CK, AST, and ALT levels can be useful serologic markers of DMD. However, profound hyperCKemia can also be seen in certain acquired or genetic neuromuscular disorders with rhabdomyolysis, such as DMD, limb–girdle muscular dystrophy, spinobulbar muscular atrophy, McArdle disease, carnitine palmitoyltransferase II deficiency, very long-chain acyl-CoA dehydrogenase deficiency, mitochondrial myopathy, dermatomyositis, and immune-mediated necrotizing myopathies [11,12,13,14,15].

The genetic diagnostic techniques for DMD have been recently gaining increasing attention [9]. NGS is a high-throughput technology that can be one of the most powerful diagnostic tools for genetic conditions [8]. It can be used as a broader, more time- and cost-effective method for identifying molecular markers of genetic diseases and neuromuscular disorders [8]. However, there are no reports of using NGS as a diagnostic tool for presymptomatic DMD in infants. NGS can be applied in infants with profound hyperCKemia to investigate the possibility of genetic diseases involving the skeletal muscle, such as DMD. In cases where these diseases are highly suspected, genetic tests, such as DES and CMA, should be performed immediately so as not to delay diagnosis.

A recent promising therapeutic approach to treat DMD is exon skipping, which is based on therapeutic attempts to restore the reading frame of the dystrophin interrupted by the premature stop of translation [16]. Skipping the target exon restores the reading frame and enables the production of a partially functional dystrophin protein rather than a nonfunctional one [17,18]. Therefore, this gene therapy aims to improve DMD symptoms to maintain the milder form of the muscular dystrophy phenotype [16,19].

The authors also provided genetic counseling to the patient’s parents to inform them about the multidisciplinary treatments, including rehabilitation, steroid therapy, and novel therapeutic agents, such as exon-skipping. The patient was transferred to the National Center for Rare Diseases in his region and is waiting for approval of genetic treatment in Korea. Therefore, we could not obtain information regarding his rehabilitation therapy and prognosis after therapeutic attempts, which was a limitation of this case report.

In this case, initial concerns of developmental delay were raised by parents, not by doctors or the Korean National Health Screening Program for Infants and Children. The Health Screening Program is available for Korean infants aged >4 months of age and aims to assess growth development using physical examination only, without laboratory tests or imaging studies. In Korea, when patients are managed at a hospital, the hospital is responsible for part of the treatment cost; the rest is paid by the National Health Insurance Service. The Health Insurance Review and Assessment Service acts as an intermediary by judging whether the request from a hospital is appropriate. If the treatment is deemed inappropriate, the hospital will not receive payment for treatment costs. The suitability of treatment is strictly assessed without considering individual situations. Several doctors tend to meet each patient as briefly as possible to save on medical costs [20]. Therefore, it may be difficult to identify diseases that require careful surveillance for diagnosis, especially those with low prevalence, such as DMD. The authors believed that AST, ALT, and CK tests should be required to verify possible disorders affecting the skeletal muscle, such as DMD. It may be difficult to make a differential diagnosis of the many diseases related to elevated AST, ALT, and CK levels in infants without an obvious weakness or motor delays. These findings would help diagnose infantile DMD in the pediatric and rehabilitation communities.

The screening for DMD has been a controversial matter for many years. Neonatal testing of CK as part of newborn screening has been explored, but it can lead to false positives; thus, the utility of screening or early diagnosis for DMD is still controversial [13]. In fact, DMD patients and their parents are strongly in favor of screening, regardless of whether the diagnosis was performed before or after symptom onset [4]. DMD patients diagnosed through newborn screening and their parents reported a high expectation and a positive impact of early diagnosis on their quality of life [4,6]. In a recent study, pilot genetic screening was conducted in male infants between 6 and 42 months with hyperCKemia [4]. However, these genetic screening tests for presymptomatic diagnosis can be problematic because of the risk of detecting intractable diseases with no treatment or prophylaxis, unexpected diseases, and variants of uncertain pathological significance. In presymptomatic diagnosis, priority should be given to the search for treatable illnesses that should not be overlooked. It is still questionable to conduct the genetic tests as the first step to save time and cost. In the present case, many doctors from the departments of pediatric infection, neurology, gastroenterology, laboratory, and rehabilitation medicine tried to diagnose the patient, and we performed many laboratory and imaging studies to determine the generally treatable illness for a considerable period time. Before implementing the DES and CMA, we minimized possible ethical issues by referring to a genetic counseling specialist. Moreover, we must consider the ethical responsibilities in genetic counseling [21]. Throughout all of the above-mentioned processes, we tried to ensure that the patient and his parents are minimally psychologically, socially, and economically burdened based on the ethical principles of genetic counseling.

In conclusion, we described a case of infantile DMD with high hyperCKemia, AST, and ALT levels in the presymptomatic state. An early diagnosis of DMD offers the opportunity to treat presymptomatic individuals and provide timely genetic counseling to families. Moreover, treatments performed as early as possible provide the greatest opportunity to slow disease progression when the least amount of muscle damage has occurred, although there is insufficient research with long-term data to date. Early presymptomatic diagnosis of DMD using the genetic diagnostic techniques should be carefully considered under an adequate psychological support system through ethical genetic counseling. This study suggests that the advanced test methods, such as NGS and CMA, are valuable techniques for accurately diagnosing infantile presymptomatic neuromuscular disorders. Further studies are required to investigate the long-term clinical outcomes in patients diagnosed at the time of no obvious symptoms of muscle weakness.

## Figures and Tables

**Figure 1 children-08-00377-f001:**
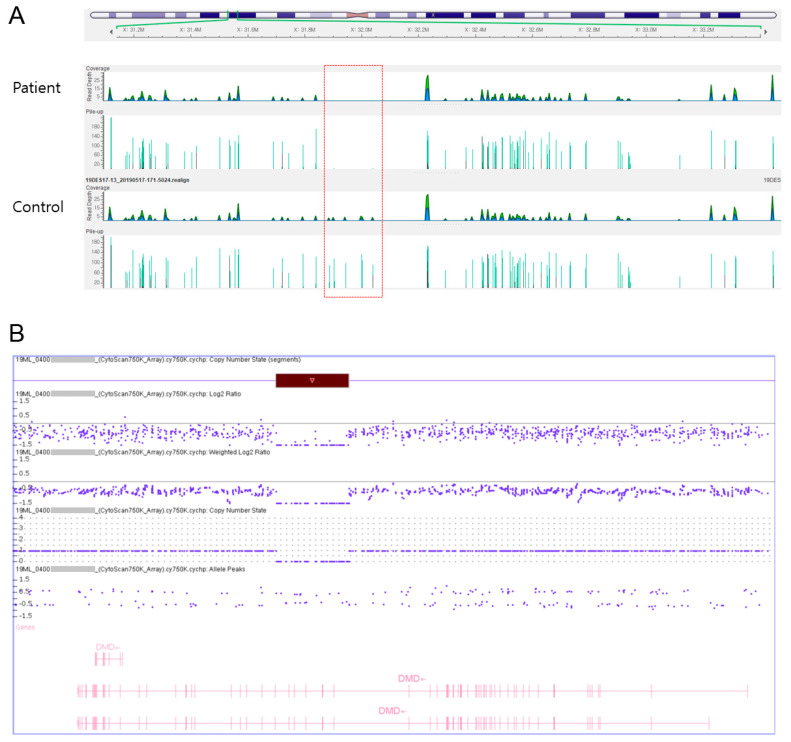
(**A**) Golden Helix GenomeBrowse visualization of hemizygous deletion in an infant with Duchenne muscular dystrophy (DMD). The X- and Y-axes represent genomic coordinates and coverage, respectively. The first row shows the patient with DMD with a hemizygous deletion of exons 45–50; the second row shows a control sample without deletion. (**B**) Microarray profile of the patient showing a hemizygous deletion of ~240 kb (31795774_32036509) in the chromosome region Xp21.1 (red bar).

## Data Availability

Not applicable.

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
