# Peer review of "Diagnosis of Duchenne Muscular Dystrophy in a Presymptomatic Infant Using Next-Generation Sequencing and Chromosomal Microarray Analysis: A Case Report"

_children, 2021, doi:10.3390/children8050377_

Round 1

Reviewer 1 Report

The authors report a 4-month-old asymptomatic boy with DMD diagnosed by DES and CMA. His growth and motor development were in the normal range. Based on the presence of high serum AST, ALT, and CK levels, they performed DES and CMA and detected the deletion of exon 45-50 in the DMD gene. The authors emphasize the benefits of early pre-symptomatic diagnosis using NGS as follows; there is a possibility of early access to new therapies such as exon skipping treatment and the advantage of early introduction of existing treatments, such as steroids and rehabilitation. However, there are serious ethical concerns with this claim.

We can justify pre-symptomatic diagnosis only when we have precise preventive and radical therapies or clear evidence that prophylactic introduction of existing therapies (non-radical therapies) can produce apparent effects.

Although novel therapeutic agents such as exon-skipping and read-through therapies are emerging in DMD, such treatment is limited to only a subset of patients. The introduction of steroids and rehabilitation at the appropriate time is essential, but there is no consensus of starting them at infancy. Early pre-symptomatic diagnosis can cause severe psychological conflicts and disturbances in the family relationship. Therefore it should be carefully considered under an adequate psychological support system.

Nonetheless, they conducted NGS from the first analysis in this patient. NGS can be problematic for pre-symptomatic diagnosis because of the risk of detecting intractable diseases having no treatment or prophylaxis, unexpected diseases, and variants of uncertain pathological significance. In pre-symptomatic diagnosis, priority should be given to the search for treatable illnesses that should not be overlooked. Then careful consideration should be given to whether or not an exhaustive analysis. It is questionable to conduct NGS from the first step to save time and cost. There have been no reports like this case so far because most doctors have ethical concerns about using NGS for pre-symptomatic diagnosis.

The ethical issues change with technological advances, social consensus, and religions.

This paper should be accepted only when authors can explain the significance of NGS as the first choice in diagnosing pre-symptomatic patients and the required procedures to resolve ethical issues. They should also refer to genetic counseling necessary to patients and family members before and after the test and the psychological and social support needed when diagnosing an intractable disease for which there is no cure.

Author Response

Response to Editor and Reviewer’s comments on Children (ISSN 2227-9067)

We appreciate and are grateful for the Editor and Reviewer’s invaluable comments and recommendations. Our responses to the Editor and Reviewers’ comments are marked in blue text, and the changes made in the revised manuscript have been indicated using line numbers. Furthermore, we have highlighted the revised parts in yellow in the revised manuscript. Moreover, we have corrected some language-related errors and improved the overall readability of the manuscript.

Response to Reviewer 1 Comments

The authors report a 4-month-old asymptomatic boy with DMD diagnosed by DES and CMA. His growth and motor development were in the normal range. Based on the presence of high serum AST, ALT, and CK levels, they performed DES and CMA and detected the deletion of exon 45-50 in the DMD gene. The authors emphasize the benefits of early pre-symptomatic diagnosis using NGS as follows; there is a possibility of early access to new therapies such as exon skipping treatment and the advantage of early introduction of existing treatments, such as steroids and rehabilitation. There are serious ethical concerns with this claim.

We can justify pre-symptomatic diagnosis only when we have precise preventive and radical therapies or clear evidence that prophylactic introduction of existing therapies (non-radical therapies) can produce apparent effects.

Although novel therapeutic agents such as exon-skipping and read-through therapies are emerging in DMD, such treatment is limited to only a subset of patients. The introduction of steroids and rehabilitation at the appropriate time is essential, but there is no consensus of starting them at infancy. Early pre-symptomatic diagnosis can cause severe psychological conflicts and disturbances in the family relationship. Therefore it should be carefully considered under an adequate psychological support system.

Nonetheless, they conducted NGS from the first analysis in this patient. NGS can be problematic for pre-symptomatic diagnosis because of the risk of detecting intractable diseases having no treatment or prophylaxis, unexpected diseases, and variants of uncertain pathological significance. In pre-symptomatic diagnosis, priority should be given to the search for treatable illnesses that should not be overlooked. Then careful consideration should be given to whether or not an exhaustive analysis. It is questionable to conduct NGS from the first step to save time and cost. There have been no reports like this case so far because most doctors have ethical concerns about using NGS for pre-symptomatic diagnosis.

The ethical issues change with technological advances, social consensus, and religions.

This paper should be accepted only when authors can explain the significance of NGS as the first choice in diagnosing pre-symptomatic patients and the required procedures to resolve ethical issues. They should also refer to genetic counseling necessary to patients and family members before and after the test and the psychological and social support needed when diagnosing an intractable disease for which there is no cure.

Response :

We appreciate the reviewer’s careful comment. We fully agree with your comment, and we would like to add an additional details that were lacking in our original manuscript.

In this case report, we would like to emphasize our diagnostic process for Duchene muscular dystrophy (DMD) in a pre-symptomatic infant, rather than the genetic diagnostic technique itself.

Before performing the genetic tests, we conducted many laboratory and imaging studies to determine the generally treatable illness for a considerable period time. Many doctors in the departments of pediatric infection, neurology, gastroenterology, laboratory, and rehabilitation medicine tried to diagnose the patient. We finally have concluded that genetic testing may be essential to identify the cause of hyperCKemia and elevated AST and ALT levels. We also provided genetic counseling to his parents to inform them about the multidisciplinary treatments, including rehabilitation, steroid therapy, and novel therapeutic agents such as exon-skipping.

It is very important to minimize possible ethical issues as you have mentioned. Given the variety of genetic tests, it is difficult to determine the most appropriate test. Therefore, we referred the case to a genetic counseling specialist of laboratory medicine. We also informed the parents about the detailed process of these genetic diagnostic techniques, and they agreed to let their child undergo such tests.

We fully agree that we must consider the ethical responsibilities required in genetic counseling (Senter, L.; Bennett, R.L.; Madeo, A.C.; Noblin, S.; Ormond, K.E.; Schneider, K.W.; Swan, K.; Virani, A.; National Society of Genetic Counselors Code of Ethics Review Task, F. National Society of Genetic Counselors Code of Ethics: Explication of 2017 Revisions. J Genet Couns 2018, 27, 9-15, doi:10.1007/s10897-017-0165-9). Throughout all of the abovementioned processes, we tried to ensure that the patients are minimally psychologically, socially and economically burdened based on of the ethical principles of genetic counseling.

In a relatively recent study, the polymerase chain reaction (PCR) test was performed to determine the cause of muscle enzyme level abnormalities in an infant, even if there were no typical symptoms observed, except for jaundice (“Russell AC, Gillis LA. Diagnosis of Muscular Dystrophy in a 6-Week-Old With Jaundice. J Pediatr Gastroenterol Nutr. 2017 Apr;64”). Recently, the use of next-generation sequencing (NGS) for diagnosing DMD has been in the spotlight ("Molecular genetic testing and diagnosis strategies for dystrophinopathiesin the era of next generation sequencing. Clinica Chimica Acta. Volume 491, April 2019, Pages 66-73"). Moreover, DMD patients and their parents are strongly in favor of screening, regardless of whether their diagnosis was performed before or after symptom onset. DMD patients diagnosed through newborn screening and their parents reported a high expectation and a positive impact of early diagnosis on their quality of life. There are no negative psycho-social impacts identified among the families with patients who were diagnosed through newborn screening (Vita, G.L.; Vita, G. Is it the right time for an infant screening for Duchenne muscular dystrophy? Neurol Sci 2020, doi:10.1007/s10072-020-04307-7.).

Therefore, if the ethical and social responsibilities of early pre-symptomatic diagnosis for DMD have been established, early diagnosis by genetic testing would be very valuable to DMD patients and their parents.

We hope that we have answered the reviewer's comments appropriately.

Below are the revisions made in our manuscript according to the reviewer’s comments.

We added a sentence describing that the use of NGS in the diagnosis of DMD is gaining much attention (line 51):

The use of NGS for diagnosing DMD also had recently been in the spotlight.

We mentioned that we checked the possibility of metabolic diseases before performing the genetic test (line 82):

The enzyme activities related to lysosomal storage diseases were also in the normal range.

We also revised the text to include that we consulted a genetic laboratory medicine specialist, before performing the NGS and CMA, to minimize any ethical problems (line 87):

Prior to the implementation of the DES and CMA, we referred the case to a genetic counseling specialist of laboratory medicine to minimize possible ethical issues. We also explained the detailed process of these genetic diagnostic techniques to the patient’s parents, before they provided written informed consent.

We add revised the text to include our interventions to minimize any ethical issues (lines 188 and 213):

Line 188

The authors also provided genetic counseling to the patient’s parents to inform them about the multidisciplinary treatments, including rehabilitation, steroid therapy, and novel therapeutic agents such as exon-skipping.

Line 213

The screening for DMD has been a controversial matter for many years. Neonatal testing of CK as part of newborn screening has been explored, but it can lead to false positives; thus, the utility of screening or early diagnosis for DMD is still controversial [13]. In fact, DMD patients and their parents are strongly in favor of screening, regardless of whether the diagnosis was performed before or after symptom onset [4]. DMD patients diagnosed through newborn screening and their parents reported a high expectation and a positive impact of early diagnosis on their quality of life [4,6]. In a recent study, pilot genetic screening was conducted in male infants aged between 6 and 42 months with hyperCKemia [4]. However, these genetic screening tests for pre-symptomatic diagnosis can be problematic because of the risk of detecting intractable diseases that have no treatment or prophylaxis, unexpected diseases, and variants of uncertain pathological significance. In pre-symptomatic diagnosis, priority should be given to the search for treatable illnesses that should not be overlooked. It is still questionable to conduct the genetic tests as the first step to save time and cost. In the present case, many doctors from the departments of pediatric infection, neurology, gastroenterology, laboratory, and rehabilitation medicine tried to diagnose the patient, and we performed many laboratory and imaging studies to determine the generally treatable illness for a considerable period time. Prior to the implementation of the DES and CMA, we minimized possible ethical issues by referring to a genetic counseling specialist. Moreover, we must consider the ethical responsibilities in genetic counseling [21]. Throughout all of the abovementioned processes, we tried to ensure that the patient and his parents are minimally psychologically, socially, and economically burdened based on the ethical principles of genetic counseling.

We also added a sentence in the conclusion that describes the importance of addressing possible ethical issues and the interventions physicians should keep in mind when prior to performing genetic tests (lines 242):

Line 242

Early pre-symptomatic diagnosis of DMD using the genetic diagnostic techniques should be carefully considered under an adequate psychological support system through ethical genetic counseling.

Reviewer 2 Report

This is a Case Report, thus an N of 1, describing a very young individual with genetic DMD that was brought to the attention of the team through an unusual route.  The authors were able to identify the subject as DMD in the absence of traditional muscle weakness through NGS and confirmed by CMA.  One a certain level, there is no real surprise, NGS and CMA are standard technologies now and would find DMD in almost all cases if the assays were used.  However, a point here is that these techniques are not typically used until a child exhibits muscle weakness as DMD does not run in families.  It seems the team was generally investigating the cause of hyper CK in the subject and found DMD, but it does highlight that early identification of DMD, prior to symptoms, is feasible and would be beneficial.  This is timely since early treatment would benefit patients and we are entering a time when multiple treatment options are available.  With the cost of NGS coming down rapidly, it may be cost effective to perform on all newborns.  It will be interesting to follow this subject if they are able to get intervention and see how much they differ on the disease track.  Regardless, as a case study this seems appropriate and I do not have any technical concerns.

Author Response

Response to Editor and Reviewer’s comments on Children (ISSN 2227-9067)

We appreciate and are grateful for the Editor and Reviewer’s invaluable comments and recommendations. Our responses to the Editor and Reviewers’ comments are marked in blue text, and the changes made in the revised manuscript have been indicated using line numbers. Furthermore, we have highlighted the revised parts in yellow in the revised manuscript. Moreover, we have corrected some language-related errors and improved the overall readability of the manuscript.

Response to Reviewer 2 Comments

This is a Case Report, thus an N of 1, describing a very young individual with genetic DMD that was brought to the attention of the team through an unusual route.  The authors were able to identify the subject as DMD in the absence of traditional muscle weakness through NGS and confirmed by CMA.  One a certain level, there is no real surprise, NGS and CMA are standard technologies now and would find DMD in almost all cases if the assays were used.  However, a point here is that these techniques are not typically used until a child exhibits muscle weakness as DMD does not run in families.  It seems the team was generally investigating the cause of hyper CK in the subject and found DMD, but it does highlight that early identification of DMD, prior to symptoms, is feasible and would be beneficial.  This is timely since early treatment would benefit patients and we are entering a time when multiple treatment options are available.  With the cost of NGS coming down rapidly, it may be cost effective to perform on all newborns.  It will be interesting to follow this subject if they are able to get intervention and see how much they differ on the disease track.  Regardless, as a case study this seems appropriate and I do not have any technical concerns.

Response :

Thank you for your encouraging comments. Despite the limitations of this article, we thank you for your feedback. As you have mentioned, we will try to conduct further research on the prognosis of this patient, compared to other patients with a similar case.

Below are the revisions made in our manuscript according to the reviewer’s comments.

We added a sentence describing that the use of NGS in the diagnosis of DMD is gaining much attention (line 51):

The use of NGS for diagnosing DMD also had recently been in the spotlight.

We mentioned that we checked the possibility of metabolic diseases before performing the genetic test (line 82):

The enzyme activities related to lysosomal storage diseases were also in the normal range.

We also revised the text to include that we consulted a genetic laboratory medicine specialist, before performing the NGS and CMA, to minimize any ethical problems (line 87):

Prior to the implementation of the DES and CMA, we referred the case to a genetic counseling specialist of laboratory medicine to minimize possible ethical issues. We also explained the detailed process of these genetic diagnostic techniques to the patient’s parents, before they provided written informed consent.

We add revised the text to include our interventions to minimize any ethical issues (lines 188 and 213):

Line 188

The authors also provided genetic counseling to the patient’s parents to inform them about the multidisciplinary treatments, including rehabilitation, steroid therapy, and novel therapeutic agents such as exon-skipping.

Line 213

The screening for DMD has been a controversial matter for many years. Neonatal testing of CK as part of newborn screening has been explored, but it can lead to false positives; thus, the utility of screening or early diagnosis for DMD is still controversial [13]. In fact, DMD patients and their parents are strongly in favor of screening, regardless of whether the diagnosis was performed before or after symptom onset [4]. DMD patients diagnosed through newborn screening and their parents reported a high expectation and a positive impact of early diagnosis on their quality of life [4,6]. In a recent study, pilot genetic screening was conducted in male infants aged between 6 and 42 months with hyperCKemia [4]. However, these genetic screening tests for pre-symptomatic diagnosis can be problematic because of the risk of detecting intractable diseases that have no treatment or prophylaxis, unexpected diseases, and variants of uncertain pathological significance. In pre-symptomatic diagnosis, priority should be given to the search for treatable illnesses that should not be overlooked. It is still questionable to conduct the genetic tests as the first step to save time and cost. In the present case, many doctors from the departments of pediatric infection, neurology, gastroenterology, laboratory, and rehabilitation medicine tried to diagnose the patient, and we performed many laboratory and imaging studies to determine the generally treatable illness for a considerable period time. Prior to the implementation of the DES and CMA, we minimized possible ethical issues by referring to a genetic counseling specialist. Moreover, we must consider the ethical responsibilities in genetic counseling [21]. Throughout all of the abovementioned processes, we tried to ensure that the patient and his parents are minimally psychologically, socially, and economically burdened based on the ethical principles of genetic counseling.

We also added a sentence in the conclusion that describes the importance of addressing possible ethical issues and the interventions physicians should keep in mind when prior to performing genetic tests (lines 242):

Line 242

Early pre-symptomatic diagnosis of DMD using the genetic diagnostic techniques should be carefully considered under an adequate psychological support system through ethical genetic counseling.

Round 2

Reviewer 1 Report

At this point, it is difficult to say that all countries and scientists have reached a consensus on the idea of when NGS should be used. Even so, the most important thing is to ensure that patients and their families have the right to make autonomous choices with careful explanation and adequate psychological support.

The additional information confirms that the authors have conducted their search carefully and with ethical considerations, and I acknowledge that my questions have been appropriately answered.